# Longitudinal Three-Dimensional Stereophotogrammetric Growth Analysis in Infants with Unilateral Cleft Lip and Palate from 3 to 12 Months of Age

**DOI:** 10.3390/jcm12206432

**Published:** 2023-10-10

**Authors:** Jennifer Kluge, Robin Bruggink, Nikolaos Pandis, Alexey Unkovskiy, Paul-Georg Jost-Brinkmann, Anne Marie Kuijpers-Jagtman, Theodosia Bartzela

**Affiliations:** 1Department of Orthodontics and Dentofacial Orthopedics, Center for Oral Health Sciences CC3, Charité—Universitätsmedizin Berlin, Corporate Member of Freie Universität Berlin and Humboldt-Universität zu Berlin, Aßmannshauser Straße 4-6, 14197 Berlin, Germany; alexey.unkovskiy@charite.de (A.U.); paul-g.jost-brinkmann@charite.de (P.-G.J.-B.); 2Radboudumc 3D Lab, Radboud Institute for Health Sciences, Radboud University Medical Center, 6500 HB Nijmegen, The Netherlands; robin.bruggink@radboudumc.nl; 3Department of Orthodontics and Dentofacial Orthopedics, School of Dental Medicine, Medical Faculty, University of Bern, Freiburgstraße 7, 3010 Bern, Switzerlanda.m.kuijpers-jagtman@umcg.nl (A.M.K.-J.); 4Department of Dental Surgery, Sechenov First Moscow State Medical University, Bolshaya Pirogovskaya Street, 19c1, Moscow 119146, Russia; 5Department of Orthodontics, University Medical Center Groningen, University of Groningen, Hanzeplein 1, 9713 GZ Groningen, The Netherlands; 6Faculty of Dentistry, Universitas Indonesia, Campus Salemba, Jalan Salemba Raya No. 4, Jakarta 10430, Indonesia; 7Department of Orthodontics, Technische Universität Dresden, Fetscherstraße 74, 01307 Dresden, Germany

**Keywords:** stereophotogrammetry, three-dimensional, cleft lip and palate, infants, facial growth

## Abstract

This longitudinal study aimed to evaluate facial growth and soft tissue changes in infants with complete unilateral cleft lip, alveolus, and palate (CUCLAP) at ages 3, 9, and 12 months. Using 3D images of 22 CUCLAP infants, average faces and distance maps for the entire face and specific regions were created. Color-coded maps highlighted more significant soft tissue changes from 3 to 9 months than from 9 to 12 months. The first interval showed substantial growth in the entire face, particularly in the forehead, eyes, lower lip, chin, and cheeks (*p* < 0.001), while the second interval exhibited no significant growth. This study provides insights into facial soft tissue growth in CUCLAP infants during critical developmental stages, emphasizing substantial improvements between 3 and 9 months, mainly in the chin, lower lip, and forehead. However, uneven growth occurred in the upper lip, philtrum, and nostrils throughout both intervals, with an overall decline in growth from 9 to 12 months. These findings underscore the dynamic nature of soft tissue growth in CUCLAP patients, highlighting the need to consider these patterns in treatment planning. Future research should explore the underlying factors and develop customized treatment interventions for enhanced facial aesthetics and function in this population.

## 1. Introduction

Orofacial Clefts (OFC) are the most common congenital craniofacial malformations worldwide. The prevalence of patients with CL/P in European countries is about 0.91 to 2.69 per 1000 live births [1]. Remarkably, the great variability in prevalence is based on racial identity, geographic region, and cleft phenotype [2]. There are various subphenotypes with an extensive severity spectrum, ranging from microforms of the cleft lip (CL), i.e., forme fruste or cleft palate (i.e., uvula bifida), to complete unilateral or bilateral cleft lip, alveolus, and palate (CUCLAP, CBCLAP) [3]. The etiologic factors are both genetic and exogenous contributors [4]. Nevertheless, genetic variants of non-cleft individuals have also been identified in non-syndromic CL/P patients [5]. Left-sided clefts are more common than right-sided clefts [6]. Genes involved in the developmental organ laterality and epigenetic factors may be responsible for cleft-sidedness [7,8]. Even though etiologic factors have been intensively studied, the pathogenic mechanism is not clearly defined [9]. Genetic studies have shown that CL/P and cleft palate only (CPO) have distinct etiologic mechanisms [10,11,12]. A strong genetic component in patients with CPO and the prevalence of this phenotype in linked malformations have been emphasized [13]. Furthermore, according to genetic association studies, the higher the phenotypic severity of orofacial cleft, the higher the prevalence of associated malformations [14,15,16]. Current studies have investigated the facial morphology [17] and nasal soft tissue symmetry [18] in young children undergoing primary surgery for CL/P repair [19] and compared the results with non-cleft individuals [20]. Genetic defects during embryological development can impact the formation of the facial skeleton and soft tissues, leading to various conditions, such as micrognathia, facial asymmetry, or orofacial clefts [21,22].

Patients with CL/P experience a lengthy treatment journey from birth, which often subjects them to the strains of societal stigmatization and social exclusion. Both facial aesthetics and functional impairments significantly impact these patients’ satisfaction and quality of life [23]. The inherent factors associated with the cleft and the surgical repair procedures can influence the growth of the maxilla and the overlying soft tissues [24]. Consequently, a longitudinal evaluation of the treatment process during this crucial period of craniofacial development is particularly interesting. Several studies have assessed craniofacial growth in CL/P patients using conventional procedures [24,25]. In recent years, various advanced methods have been employed for craniofacial diagnosis and treatment planning, enhancing the management of patients with CL/P. These include 3D and temporal (4D) stereophotogrammetry, 3D digital models, and cone beam computerized tomography (CBCT), which have emerged as valuable adjunct tools in this context [26,27]. In particular, 3D imaging has enabled a comprehensive evaluation of craniofacial morphology and growth, facial symmetry and aesthetics, and bone preservation on a longitudinal basis. This technology has proven to be a reliable method for virtual treatment planning and treatment outcome evaluation of individuals with CL/P [28,29].

Stereophotogrammetry is a 3D reconstructional imaging technique that captures the facial structures within a short acquisition time [17]. It is based on a conventional digital camera setting. Over the last years, 2D and 3D photography in anthropometric evaluation with advanced computer software data analysis have become a significant advantage in soft tissue and growth analysis, thereby expanding clinical practice [30]. Three-dimensional imaging uses conventional photography in a fixed base distance to assess cleft structures, soft tissue symmetry, and morphological analysis without ionizing radiation exposure. Data on adult patients or older children with CL/P are available [31,32,33], while such data on young children are scarce [22].

Current studies have investigated facial morphology [17] and nasal soft tissue symmetry [18] in young children undergoing primary surgery for CL/P repair [19] and compared the results with non-cleft individuals [20].

The function, growth, type, and timing of the surgical procedures correlate with and influence these patients’ facial soft tissue and skeletal balance. The lack of consensus concerning the surgical protocol, the adaptation of treatment procedures based on the expected prognosis, and the surgeons’ preference and experience make an evidence-based choice for a certain surgical procedure in neonates challenging [25,34]. Hence, the primary goal of this study was to assess how the facial and nasolabial regions of children with CUCLAP change over time at the ages of 3, 9, and 12 months after their initial surgical interventions. This evaluation focused on understanding how facial growth, factors related to the surgery itself, and inherent cleft-related parameters impact the facial development.

## 2. Materials and Methods

### 2.1. Participants

The present study was a longitudinal 3D facial growth assessment of children with CUCLAP during their first year of life. All examinations and surgical interventions were conducted by the same team of surgeons and based on the surgical protocol of patients with CUCLAP of the Charité—Universitätsmedizin Berlin. Three-dimensional stereophotogrammetric photos were acquired for 22 CUCLAP patients as standardized treatment documentation at 3, 9, and 12 months, according to the recommendations of the Eurocleft Project [35]. The inclusion criteria of this study were Caucasian, non-syndromic children with CUCLAP. All surgical procedures were planned from birth onwards in interdisciplinary cooperation with other departments of the same center. Children with associated malformations, syndromes, or just a Simonart’s band were excluded from this study. Patients’ recruitment was performed from November 2016 to August 2019. The patients’ parents gave informed consent for their inclusion in this study. This study was a single-center study at Charité—Universitätsmedizin Berlin, and it was approved by the institution’s medical Ethics Committee (EA2/096/16).

### 2.2. Surgical Protocol

Since 1996, the same surgical protocol has been employed at Charité Campus Virchow, Cleft Palate Craniofacial Unit, Berlin. The treatment protocol is briefly summarized in Table 1. In addition to the surgical procedures, passive infant orthopedic intraoral appliances were used in all children. At about three months of age, lip adhesion or a gingivoperiosteoplasty and soft palate closure were employed in one step. The hard palate and lip were repaired with Millard’s technique at 9 months. The data were acquired before lip adhesion at three months and after hard palate and lip closure at 9 and 12 months of the patient’s age, respectively (Table 1).

### 2.3. Data Acquisition

The same trained researcher (JK) took the 3D facial photographs. The 3D stereophotogrammetric camera and the manufacturer’s software program Modular System v1.0 (3dMDface^TM^ System, 3dMd Ltd., Atlanta, GA, USA) were used. The installed camera system comprises two pods containing three digital cameras each and a flash module. The image capturing duration was 1.5 ms, resulting in a high-resolution 3D image with around 40,000 polygons per square inch. On average, three or four 3D photographs were taken for each child on each occasion, depending on the infants’ cooperation [36]. Preparation was performed following a strict protocol. The patient was upright at a fixed equilateral distance in front of the system. The parents stabilized the child’s head in a neutral position and kept hair and clothes away from the face. The lighting was standardized using the same lights and brightness during this study. System calibration was conducted prior to every session. After the acquisition, all images were directly examined to see if they met the following criteria: (1) natural head position, (2) relaxed facial expression, open eyes, and lips in contact without strain, (3) no hair on the forehead and cropped hair or clothes from the head or neck, (4) both ears or at least one ear wholly captured, and (5) no data gaps in the reference points.

A second critical assessment was made later to select the appropriate images for further processing. All files were imported into the Geomagic freeform software (3D Systems, Rock Hill, SC, USA) as a mesh object without texture to allow recording in most 3D software and to downsize the recorded data. The background and neck of the included photographs were cropped. The facial soft tissue reference frames were constructed as described by Brons et al. [37]. All images with a cleft on the right side were mirrored on the sagittal plane, and faces with a left-sided cleft were created. Next, all CUCLAP photos were imported into 3DMedX^®^ software Version 1.2.27.1 (3D Lab Radboudumc, Nijmegen, The Netherlands). Only cropped images without skin texture were selected [17].

Before average faces could be reconstructed, each image was standardized using the ‘Mesh Monk’ library [38] using a standardized template. The used template consists of 7160 evenly spread datapoints manually annotated to enable regional analysis. The regions include the forehead, nose, eyes, upper lip, lower lip, cheeks, and chin (Figure 1). The standardization process, as described in a previous study by Bruggink et al. [39], can be summarized in the following steps:(1)A predefined, evenly spread template was roughly aligned on the facial model of the patient using a procrustus technique with both endocanthia, cheilon, and the pronasale landmark as reference points.(2)A rigid registration technique further aligned the template.(3)A non-rigid registration technique described by Matthews et al. [38] was used to morph every individual point on the template towards the contours of the face.(4)Finally, each point of the template was projected onto the facial mesh. This results in a standardized, uniform indexed mesh in which individual datapoints correspond with those of other standardized meshes.

Average faces were generated for 3 (T1), 9 (T2), and 12 months (T3) of age. The comparison of the average faces at the defined time points (T1 to T2 and T2 to T3) was visualized by color-coded maps.

**Figure 1 jcm-12-06432-f001:**
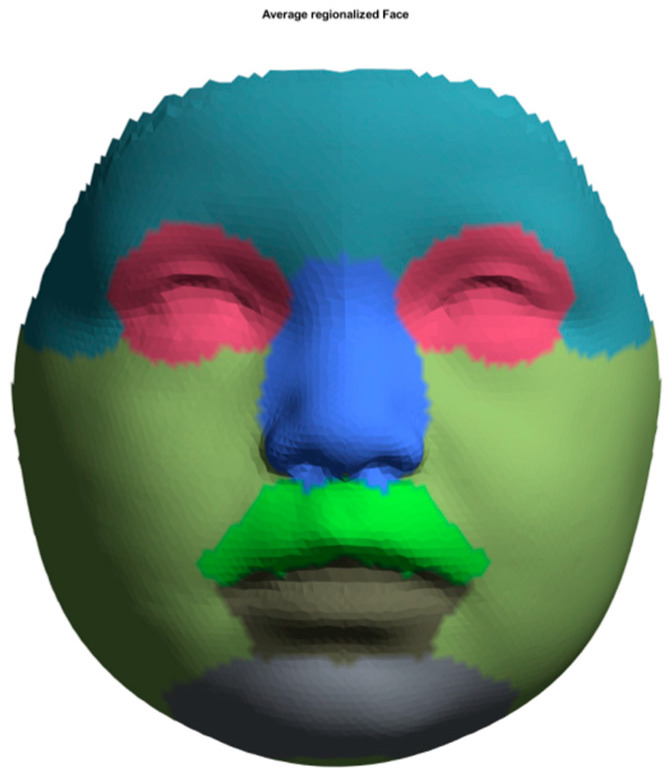
Selected areas for assessing facial development and growth: forehead (light steel blue), eyes (rose), nose (royal blue), upper lip and subnasal (lime green), lower lip (anthracite), cheeks (olive green), and chin (grey).

### 2.4. Statistical Analysis

Distance maps were created to evaluate the growth of the patients’ faces at different time points. The color-coded maps visualized the growth increments from 3 to 9 months (T1–T2) and from 9 to 12 months (T2–T3). The set minimum and maximum range was between −5 mm and 5 mm, and the standard deviation (std), 90th percentile, and 95th percentile were calculated. The intra-observer reliability was tested by ten randomly selected patient images. Paired sample *t*-test was applied using the software SPSS Statistics 25.0 (IBM^®^, Ehmingen, Germany). The statistical significance was set at *p* < 0.05.

## 3. Results

### 3.1. Sample and Image Selection

Initially, 40 babies with CL/P were included. After the first examination, 18 children were excluded because they did not meet the inclusion criteria, moved to another city, or had an associated syndrome. In total, 164 available images of 22 CUCLAP patients were assessed, and all duplicate images were excluded. After a detailed examination, more 3D photographs were excluded (*n* = 81) because they did not meet the timepoint criteria initially defined or showed involuntary facial expressions. Table 2 provides an overview. Our study was an observational study in which we followed longitudinally one group of patients (CUCLAP; *n* = 22). The number of included patients was not based on a sample size calculation but on the availability of patients meeting the inclusion criteria. Although a priori, we had confidence that this number would be big enough to warrant this study; a lack of power, especially in cases where no statistically significant effects were found, can complicate the interpretation of the findings in this study. Therefore, for all comparisons, we gave not only estimates of effects and *p*-values but 95% confidence intervals as well, so the reader can easily see the precision of the estimates produced from each test performed within this study. The null hypothesis for this study posits that there is asymmetric facial growth in children with CUCLAP, particularly in the nasolabial affected area. Furthermore, it suggests significant differences in the growth patterns observed in the entire face and specific facial regions between the periods spanning from 3 to 9 months and from 9 to 12 months of age.

### 3.2. Average Faces of CUCLAP

Based on the superimposition, normative average faces have been created for infants at 3, 9, and 12 months (T1, T2, and T3, respectively), representing the lip and soft and hard palate post-surgical state (Table 1). All pre-operative photos (T0) were excluded (*n* = 84) because they did not match the inclusion criteria defined. Only 9 of the 22 patients were eligible in the T0 population. All images were superimposed using the described reference frames in all three dimensions.

### 3.3. Assessment of Facial Growth

The superimposed images at T1 showed growth in the forehead and right nostril region (non-cleft side). The smallest inter-surface distance values were in the eye’s area and the affected part of the lip and nose. The nose and upper lip asymmetry were visible. The nose tip pointed towards the unaffected side. Right after lip adhesion, the affected side’s upper lip was located anteriorly compared to the non-affected side. Partial growth impairment in the upper lip region was observed from T1 to T2 (mean: 1.92 mm, std: 2.45 mm). The upper lip showed asymmetrical values. The asymmetry of the nose with a mean of 1.05 mm (std: 2.28 mm) was also clearly visible. The tip of the nose remained towards the unaffected side of the cleft and showed more growth than the rest of the nose at T1.

Moreover, asymmetry of the upper lip was visible. Right after lip adhesion, the affected part of the upper lip was located anteriorly compared to the area of the closed upper lip tissues. The growth of the affected side of the CUCLAP faces seemed less pronounced. The chin showed the biggest increase, with a mean of 3.17 mm (std: 0.29 mm). All results are displayed in Table 3.

At 9 months of age, the surface values of the upper facial half were still very high and similar to the 3-month-old children with CUCLAP. The values in the upper and lower lip of the right side increased, and the upper lip symmetry was attained (Table 3). Furthermore, the area of the nostril of the affected side showed an asymmetry with lower values of growth changes compared to the non-affected side.

Compared to the first interval, the superimposition of the 3D photos at T2 and T3 showed an even distribution of the growth all over the face with a 1.98 mm absolute (abs) mean. Moreover, there is further increase on the forehead (mean: 1.35 mm, std: 1.11 mm) and the chin (mean: 2.30 mm, std: 0.34 mm). The growth in the upper lip region was limited to 0.65 mm (mean: 0.65 mm, std: 2.77 mm). At 12 months, the superimposed images showed symmetry improvement and smaller changes in the chin and mandible.

### 3.4. Growth of the Face in Total and Facial Areas from 3 to 9 Months of Age

Table 3 presents the results of the superimposed average faces from 3 to 9 months (T1 to T2). The values of the entire face and the selected areas are presented. The mean facial growth was 2.31 mm at the interval of T1 to T2. Regarding the selected areas, the largest changes were observed in the chin and upper lip regions (mean: 3.17 mm and 1.92 mm, respectively) during the first interval (T1 to T2). The standard deviation (std) from T1 to T2 ranged from 0.29 mm (chin) to 2.45 mm (upper lip).

In the interval T1–T2, statistically significant growth was observed in the nose (*p* < 0.05) and the upper lip (*p* < 0.05). Furthermore, the areas of the forehead, eyes, lower lip, chin, and cheeks showed statistically significant mean changes (*p* < 0.001) (Table 3).

Figure 2A shows the distance kit of the babies (T1 to T2), illustrating the facial changes by superimposing the 3D photos. The green shades represent the positive and the red the negative scale values. Color intensity is associated with more extreme values. The same visualization is shown in Figure 3A for the interval of T2 to T3. Additionally, Figure 2B shows the superimposition of the average face at T1 (mint green) and the average face at T2 (dark blue) for simplification. The superimposition of the average face at T2 (dark blue) and the average face at T3 (purple) is illustrated in Figure 3B. The superimposition of the average faces of the respective time intervals in Figure 2B and Figure 3B allows for an immediate assessment of the areas of the face that are growing more strongly or less strongly and enables a clear visualization.

### 3.5. Growth of the Face in Total and Facial Areas from 9 to 12 Months of Age

The overall facial soft tissue surface’s mean growth was 1.98 mm at the interval of T2 to T3, as shown in Table 4. Regarding the selected areas, the greatest increase occurred in the upper lip (mean: 0.65 mm, std: 2.77 mm) and the chin (mean: 2.30 mm, std: 0.34 mm) during the second interval (T2 to T3). The highest standard deviation from T2 to T3 is noticed in the upper lip, with 2.77 mm, indicating a greater variation in these regions of the faces compared to T1 to T2. The lowest standard deviation is noticed in the chin area, with 0.34 mm. During the second age period (T2 to T3), the growth seems more balanced all over the face. The areas of the upper lip (*p* = 0.2834) and the nose (*p* = 0.1221) showed no statistically significant facial changes. The forehead, eyes, lower lip, and chin area showed statistically significant differences (*p* < 0.001).

Moreover, the values for the full face and the region of the cheeks showed significant differences (*p* < 0.05). Table 4 presents the statistical analysis of the superimposed average faces from 9 to 12 months of age (T2 to T3). It shows the values of the full face and the determined areas.

## 4. Discussion

In this study, the null hypothesis, that there is asymmetric facial development after primary cleft repair surgical procedures, was partially confirmed. The findings revealed that there were significant improvements in facial soft tissue growth between 3 and 9 months, particularly in the regions of the chin, lower lip, and forehead. However, an uneven growth was observed in the upper lip, philtrum, and nostrils throughout both time intervals, with an overall decline in growth from 9 to 12 months. This indicates that while there were significant differences in certain facial areas, there were no significant differences in others, aligning with the null hypothesis for those specific regions. These results underscore the complexity of soft tissue growth in CUCLAP patients and emphasize the need for a nuanced interpretation when considering treatment planning. This cohort exhibited significantly greater growth in the upper lip area during the first interval (T1–T2) from 3 to 9 months of age. Due to the lip adhesion, the color-coded map showed improvement in this area as well as in the chin and forehead. At nine months of age, the hard palate underwent surgery. Distance maps indicated no significant growth values during the second interval (T2–T3) for the upper lip and the nose. However, more growth was observed in the nose and upper lip, albeit with irregular distribution.

A noticeable increase in growth occurred within the region encompassing the nasal tip and nostrils, accompanied by concurrent improvements in symmetry. Despite certain limitations of this study, such as the sample size, it offers a prospective longitudinal assessment with a rapid and straightforward data acquisition technique. We present preliminary observations of patients with unilateral clefting. Intrinsic and iatrogenic factors affect the growth patterns in patients with cleft. We can confirm the limpness of the nostril rim on the cleft side. The individuals in this study were treated by the same team of surgeons and the same treatment protocol based on the individual’s conditions. The second interval exhibited the highest standard deviation in the upper lip (std: 2.77 mm) and nose (std: 1.96 mm) regions, indicating significant data heterogeneity. This wide range of measurement data in this area corroborates findings in the literature by Al-Rudainy et al. [40].

Nowadays, sophisticated 3D techniques have replaced 2D imaging. Facial morphology is captured using CBCT or conventional computed tomography (CT), different laser scanning systems, 3D magnetic resonance images, and 3D stereophotogrammetry devices [25,41,42,43]. The 3D stereophotogrammetry is a valuable tool for evaluating longitudinal events occurring during craniofacial growth and development in individuals with craniofacial deformities. It provides a fast, safe, and easy way to capture images. Most 2D and 3D imaging techniques encompass a method error due to the measurement system, type of landmark, and the observer [24]. Studies like this one evaluate infants with CUCLAP for a year after the initial surgical interventions [44]. However, most studies deal with patients older than three years or pictures that have been taken under general anesthesia [45,46]. Furthermore, palatal growth has also been evaluated on dental casts in infants with CLAP [47,48].

Sexual dimorphism was not addressed in this study. During the first year of life, sex differences mainly related to weight and body measurements have been confirmed [49,50]. However, the study conducted by Matthews et al. revealed the presence of sexual craniofacial dimorphism, to a certain extent, as early as one year of age. Frontal and intercanthal width and nasal and labial measures showed sexual dimorphism only in the transition to puberty. In adulthood, male and female facial shapes are clearly distinguished [50].

Furthermore, developmental heterogeneity is expected in patients with an orofacial cleft after the surgical intervention. Infants with CUCLAP with facial soft tissue asymmetries were observed until four years of age [40]. Asymmetries were significantly diminished after the primary lip repair. Nevertheless, residual asymmetry in the nasolabial and upper lip areas were observed [40]. Longitudinal evaluation is required until craniofacial growth ceases to yield conclusive results. Therefore, we followed our CUCLAP patients and evaluated facial growth modifications in total and selected areas of infants’ faces with CUCLAP at 3, 9, and 12 months of age using 3D stereophotogrammetry.

This study employed a reference frame technique for infant growth assessment [37]. Growth analysis was based on morphometric changes [36,39]. The image setup using the children’s reference frame technique described by Brons et al. [37] is comparable to registration on the cranial base. This method facilitates and standardizes the 3D longitudinal treatment outcome evaluation [37]. An average face template was superimposed. Furthermore, the regions were automatically selected with a template to reduce the reproducibility error. A comparison between CUCLAP patients and controls was presented in a previous study [17].

In our study, we used the 3dMD system, which is costly and not readily available in low- and low–middle-income countries. An interesting and perhaps trendsetting tool is the use of porTable 3D stereophotogrammetry devices. Ritschl et al. investigated the significance of low-budget and porTable 3D stereophotogrammetry [51]. The purpose was to represent the nasal area in plaster models with another system because most 3D stereophotogrammetry systems are costly and only for stationary utilization. Moreover, the VECTRA^®^ system or the FUEL3D^®^ SCANIFY^®^ system scanner was mentioned as an alternative method for soft tissue analysis, even in newborns [52,53].

Another study examining facial soft tissues using a face scan method on smartphones showed less accuracy than stereophotogrammetry [54]. A new approach to quantifying facial asymmetry in patients with CUCLAP during function is 4D imaging (video stereophotogrammetry) [55]. Four-dimensional imaging is an assessment tool for the dynamic facial muscles’ alterations of asymmetric motion paths in patients with craniofacial anomalies or after orthognathic surgery.

Further studies should address early identification and primary prevention in OFC, e.g., interventions to minimize risk factors, including genetic counseling [56]. Other factors related to the growth of patients with CUCLAP, such as sexual dimorphism, do not play a role in early childhood [49], which is why this aspect was not considered in this study.

Longitudinal data in young children with a representation of color-coded maps are rare due to methodological challenges. The average face of the evaluated group of patients should be constructed for group or time point comparisons. The color-distance maps ensure a fast overview of the full facial growth changes and separate regions. Published normative data presented rapid growth in the forehead and chin during the first year of life [36]. The brain growth spurt and the postnatal mandibular growth have been observed in the first six months of life [57], with fewer changes from 9 to 12 months of age [36].

Most studies on that topic suffer from a low sample size, which is essential to obtain valuable data. We could not collect enough 3D images of newborns from birth to three months of age (T0 to T1) due to cooperation challenges, including communication barriers. Engaging children in photographic data acquisition during their first year of life is demanding, especially after long clinic appointments with the other cleft team disciplines. Moreover, the parents or a trained research participant must support the infant’s head. The facial expression should not be underestimated [58], especially when assessing the nasolabial area of infants with CL/P. A trained team of photographers should be involved in these demanding registrations. In longitudinal studies, involuntary facial expressions could influence the reliability of calculated growth increments [58,59]. The primary goal of this study was to assess how the facial and nasolabial regions of growing children change over time at the ages of 3, 9, and 12 months after their initial surgical interventions.

Further investigations should aim to compare this with a non-cleft sample within the same age group or a control group from a different ethnicity, if available. Our evaluation focused on understanding how facial growth and factors related to the surgery itself and inherent cleft-related factors impact these changes. So, all observed changes are a combination of growth and treatment effects, and it is not possible to differentiate between the two. Further limitations, such as hearing loss related to the condition, could affect a child’s collaboration at older ages.

Moreover, it is crucial to follow a precise operation protocol because deviations from the registration time points could also influence the reliability of the results. In this case, it allows for an early evaluation of a certain treatment protocol. For example, the treatment protocol employed in this center is rather extensive and time consuming compared to other treatment protocols. By comparing early results of treatment protocols from different centers with this non-invasive method, it may be possible to determine early differences; hence, treatment protocols could be adapted.

In this study, only infants of Caucasian origin were included, limiting this study’s sample size. Another limitation is the absence of a control group of non-affected normal-growing individuals. Further studies should also consider craniofacial growth in non-affected and affected individuals from a distinct racial background until growth cessation. It is important to note that a larger sample size is essential for studies aiming to create mean facial representations.

The learning curve of the researchers and the surgical team must be considered when evaluating the treatment outcome. The skills, experience, and a well-planned study design might influence the outcome [56].

Considering the factors above, it is imperative to undertake longitudinal cohort studies with a substantial sample size, despite the associated costs and time-intensive nature, to obtain meaningful results. Such an endeavor appears feasible only through collaborative efforts within international research consortia [60]. Additionally, long-term monitoring is needed while investigating cleft patients, because late and rare treatment outcomes should be evaluated.

## 5. Conclusions

The findings of this study showed significantly improved facial soft tissue growth from 3 to 9 months of age in children with a CUCLAP. From 9 to 12 months of age, growth decreased. The color-coded maps displayed growth all over the face and in the nasolabial region. However, growth distribution was uneven in the operated region of the face during the observational time points. In summary, even growth was noticed in the chin, lower lip, and forehead area. During both intervals, less growth was recorded in the upper vermillion border, the philtrum, and the nostrils.

In conclusion, the results of this study provide valuable insights into the facial soft tissue growth patterns of children with a CUCLAP during their critical developmental months. Our findings highlight a significant improvement in facial soft tissue morphology from 3 to 9 months of age, particularly in the chin, lower lip, and forehead areas. However, it is essential to note that this growth was not evenly distributed across the entire facial region, with less growth observed in the upper vermillion border, philtrum, and nostrils during both intervals. Furthermore, from 9 to 12 months of age, there was a decrease in overall growth. These color-coded maps clearly illustrate the dynamic nature of soft tissue growth in these patients. It is crucial for clinicians and researchers to consider these growth patterns when assessing and planning interventions for children with CUCLAP. Further research is warranted to better understand the factors contributing to the uneven growth distribution and to develop tailored treatment approaches to optimize facial aesthetics and function in this population.

## Figures and Tables

**Figure 2 jcm-12-06432-f002:**
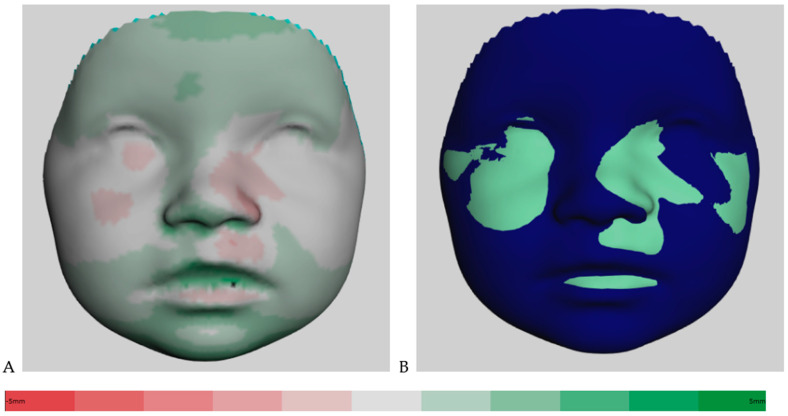
(**A**) Color-coded distance map of the superimposed average faces from 3 to 9 months (T1 to T2), ranging from red (−5 mm) to green (5 mm). (**B**) Visualization of superimposed average faces showing only positive or negative values.

**Figure 3 jcm-12-06432-f003:**
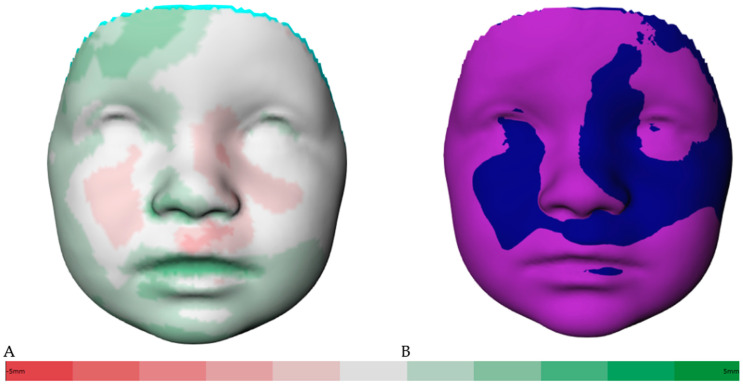
(**A**) Color-coded distance map of the superimposed average faces from 9 to 12 months (T2 to T3), ranging from red (−5 mm) to green (5 mm). (**B**) Visualization of superimposed average faces showing only positive or negative values.

**Table 1 jcm-12-06432-t001:** Surgical and infant orthopedic protocol for CLP treatment during the first year of life at Charité—Universitätsmedizin Berlin.

Age	Treatment Protocol
Prenatal to 3 months	Parent counseling Neonatal presurgical infant orthopedics
2 months	Nasoalveolar molding, Latham’s appliance (for jaw segment relocation, four weeks before surgery)
3–4 months	Lip adhesion, soft palate closure (Kriens’ technique), gingivoperiosteoplasty
9 months	Hard palate closure, lip-nose-plasty (Millard’s technique)
12 months	Lip-nose-plasty (Millard’s technique), if not performed previously

**Table 2 jcm-12-06432-t002:** Sample and image selection of all CUCLAP subjects.

Age	0 Months, T0	3 Months, T1	9 Months, T2	12 Months, T3	Total
Available infants with CL/P	40	40	38	36	26
Included infants with CUCLAP	22	22	22	22	22
Included images after a detailed assessment	9	22	15	19	83

**Table 3 jcm-12-06432-t003:** Changes of the full face and the selected facial areas of superimposed average faces from 3 to 9 months (T1–T2).

Region	Min (mm)	Mean Growth Difference Range (mm)	Median (50%)	Max (mm)	5% CI (mm)	95% CI (mm)	Std (mm)	Abs Mean Growth (mm)	Abs Std (mm)	*p*-Value
Full face	−3.50	1.89	2.23	4.09	−2.16	3.36	1.48	2.31	0.65	<0.001 **
Nose	−3.12	1.05	2.24	3.14	−2.57	3.06	2.28	2.48	0.38	0.0416 *
Forehead	−1.05	1.65	1.74	2.26	0.89	2.12	0.43	1.66	0.38	<0.001 **
Eyes	−1.98	1.05	1.36	1.98	−1.85	1.91	1.15	1.53	0.32	<0.001 **
Upper lip	−3.47	1.92	2.82	4.09	−3.28	3.79	2.45	3.07	0.45	0.0014 *
Lower lip	−3.50	2.30	2.85	4.04	−3.20	3.29	1.76	2.88	0.27	<0.001 **
Chin	2.54	3.17	3.13	3.81	2.77	3.71	0.29	3.17	0.29	<0.001 **
Cheeks	−2.91	1.94	2.35	3.98	−2.11	3.06	1.39	2.33	0.49	<0.001 **

CI = confidence interval; Std = standard deviation; Abs mean = absolute mean; Abs std = absolute standard deviation; * *p* < 0.05; ** *p* < 0.001.

**Table 4 jcm-12-06432-t004:** Changes of the full face and the selected facial areas of superimposed average faces from 9 to 12 months (T2–T3).

Region	Min (mm)	Mean Growth Difference Range (mm)	Median (50%)	Max (mm)	5% CI (mm)	95% CI (mm)	Std (mm)	Abs Mean Growth (mm)	Abs Std (mm)	*p*-Value
Full face	−3.45	1.20	1.96	3.48	−2.15	2.68	1.64	1.98	0.48	0.0025 *
Nose	−2.79	0.67	1.85	2.90	−1.87	2.64	1.96	2.05	0.32	0.1221
Forehead	−1.64	1.35	1.86	2.45	−1.06	2.30	1.11	1.67	0.52	<0.001 **
Eyes	−1.97	1.04	1.54	2.13	−1.72	2.07	1.24	1.57	0.40	0.0007 **
Upper lip	−3.45	0.65	2.40	3.41	−3.36	3.22	2.77	2.82	0.34	0.2834
Lower lip	−2.53	2.05	2.05	3.39	1.60	2.90	0.64	2.11	0.41	<0.001 **
Chin	1.65	2.30	2.25	3.29	1.80	2.97	0.34	2.30	0.34	<0.001 **
Cheeks	−2.73	0.97	1.93	3.48	−2.41	2.48	1.78	1.99	0.38	0.0185 *

CI = confidence interval; Std = standard deviation; Abs mean = absolute mean; Abs std = absolute standard deviation; * *p* < 0.05; ** *p* < 0.001.

## Data Availability

Not applicable.

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
