# Peer review of "Longitudinal Three-Dimensional Stereophotogrammetric Growth Analysis in Infants with Unilateral Cleft Lip and Palate from 3 to 12 Months of Age"

_jcm, 2023, doi:10.3390/jcm12206432_

Round 1

Reviewer 1 Report

Dear colleagues!

Thanks for your research.

I have a number of questions essentially

1. What was the null hypothesis?

2. Write how the sample size was calculated

3. Detail image 2: the difference is not particularly clear

4. Don’t you think that the t-test alone is not enough for reliable results? Why was there such statistical processing?

5. In the introduction, little attention is paid to the growth of the facial skeleton; it needs to be supplemented

Author Response

Dear Reviewer,

attached you will find the point-by-point response to the comments. Please see the attachment below.

Yours sincerely,

Jennifer Kluge

Reviewer 2 Report

Dear editor and author,

This is a well-prepared study which aims to aim to evaluate facial growth and soft tissue changes in infants with complete unilateral cleft lip, alveolus, and palate (CUCLAP) at ages 3, 9, and 12 22

months..  It presents a normative method and adequate processing of the data for a definite outcome that underscore the dynamic nature of soft tissue growth in CUCLAP patients, highlighting the need to consider these patterns in treatment planning. Future research should explore underlying factors and develop customized treatment interventions for enhanced facial aesthetics and function in this population. Here are some suggestions:

Comments 1:

Does the author calculate the sample size? Otherwise, the statistics are not conceivable.

Comments 2:

This study only included CUCLAP patients, however, to illuminate the growth rate of CUCLAP people, I highly suggest the author add another healthy infant as the control group, therefore, they can investigate the difference between CUCLAP and healthy infants during facial growth.

Comments 3:

The facial growth may also be influenced by other factors such as gender, nutrient, do the author consider these factors?
Comments 4:

How will this study pave the way for oral and maxillofacial surgeons to diagnose, and make treatment plans for CUCLAP patients?

Dear editor and author,

This is a well-prepared study which aims to aim to evaluate facial growth and soft tissue changes in infants with complete unilateral cleft lip, alveolus, and palate (CUCLAP) at ages 3, 9, and 12 22

months..  It presents a normative method and adequate processing of the data for a definite outcome that underscore the dynamic nature of soft tissue growth in CUCLAP patients, highlighting the need to consider these patterns in treatment planning. Future research should explore underlying factors and develop customized treatment interventions for enhanced facial aesthetics and function in this population. Here are some suggestions:

Comments 1:

Does the author calculate the sample size? Otherwise, the statistics are not conceivable.

Comments 2:

This study only included CUCLAP patients, however, to illuminate the growth rate of CUCLAP people, I highly suggest the author add another healthy infant as the control group, therefore, they can investigate the difference between CUCLAP and healthy infants during facial growth.

Comments 3:

The facial growth may also be influenced by other factors such as gender, nutrient, do the author consider these factors?
Comments 4:

How will this study pave the way for oral and maxillofacial surgeons to diagnose, and make treatment plans for CUCLAP patients?

Author Response

Dear Reviewer,

Please see the attachment below with the point-by-point response to the comments.

Thank you very much

Yours sincerely,

Jennifer Kluge
